# RETRACTED: Characterization and Antibacterial Activity of 7S and 11S Globulins Isolated from Cowpea Seed Protein

**DOI:** 10.3390/molecules24061082

**Published:** 2019-03-19

**Authors:** Seham Abdel-Shafi, Abdul-Raouf Al-Mohammadi, Ali Osman, Gamal Enan, Samar Abdel-Hameid, Mahmoud Sitohy

**Affiliations:** 1Botany Department, Faculty of Science, Zagazig University, Zagazig 44519, Egypt; gamalenan@ymail.com (G.E.); dr.samar19891989@gmail.com (S.A.-H.); 2Department of Science, King Khalid Military Academy, P.O. Box 22140, Riyadh 11495, Saudi Arabia; almohammadi26@hotmail.com; 3Biochemistry Department, Faculty of Agriculture, Zagazig University, Zagazig 44511, Egypt; ali_khalil2006@yahoo.com (A.O.); mzsitohy@hotmail.com (M.S.)

**Keywords:** cowpea protein, 7S globulin, 11S globulin, antibacterial activity, meat

## Abstract

The present work was carried out to determine the characteristics and antibacterial activity of 7S and 11S globulins isolated from cowpea seed (*Vigna unguiculata* (L.) Walp.). The molecular mass of 7S globulin was demonstrated by SDS-PAGE bands to be of about 132, 129 and 95 kDa corresponding the α^/^, α and β subunits. The molecular mass of 11S globulin was demonstrated by SDS-PAGE bands to be existed between 28 and 52 kDa corresponding the basic and acidic subunits. The minimum inhibitory concentrations MICs of 7S and 11S globulins isolated from cowpea seed were determined against Gram positive bacteria viz: *Listeria monocytogenes* LMG 10470, *Listeria ivanovii* FLB 12, *Staphylococcus aureus* ATCC 25923 and *Streptococcus pyogenes* ATCC 19615, and Gram negative bacteria such as *Klebsiella pneumonia* ATCC 43816, *Pseudomonas aeruginosa* ATCC 26853, *Escherichia coli* ATCC 25922 and *Salmonella* ATCC 14028 using disc diffusion assay; they were showed to be in the range 10 to 200 µg/mL. Transmission electron microscope (TEM) examination of the protein-treated bacteria showed the antibacterial action of 11S globulin against *S. typhimurium* and *P. aeruginosa* was manifested by signs of cellular deformation, partial and complete lysis of cell components. Adding 11S globulin at both concentrations 50 and 100 µg/g to minced meat showed considerable decreases in bacterial counts of viable bacteria, psychrotrophs and coliforms compared to controls during 15 days storage at 4 °C, reflecting a promising perspective to use such globulin as a meat bio-preservative.

## 1. Introduction

The unbalanced consumption and overuse of antimicrobials have led to the growth of antibiotic-resistant bacteria and this has forced scientists to search for novel antimicrobial agents with different mechanisms of action to be used as food preservatives or in treatment of human diseases [1,2,3,4]. Cationic antimicrobial peptides or proteins (AMPs) are still the best choice and most promising candidates. Numerous studies have shown the have broad spectrum of antimicrobial activities of many natural AMPs against Gram-positive and Gram-negative pathogenic bacteria [5,6,7,8,9,10,11].

Microbial contamination of food by food-borne pathogens is one of the main problems leading to reduced food shelf life and possible food poisoning, triggering major health hazards. Many chemical additives are, therefore, used as food preservatives to promote food safety and extend food shelf life. The potential harmful effects associated with using chemical preservatives have switched on the search for safe natural bio-preservatives with considerable antimicrobial capacity, to be used in food industries [12,13,14,15,16,17,18,19,20].

Cowpea [*Vigna unguiculata* (L.) Walp] is a legume seed of African origin grown mainly in tropical and subtropical regions and widely distributed throughout the world. It represents an important source of protein for the world population, especially in the lower income segments [21]. Legume globulins constitute the majority of soybean seed proteins and can be subdivided into two main types according to their sedimentation coefficients: 7S and 11S globulins. The 11S fraction has a molecular mass of about 350 kDa and is composed of acidic (37–42 kDa) and a basic polypeptide (20 kDa) subunits, linked together by a disulphide bond [22].

Globulins represent the majority of cowpea seed proteins and constitute over 51% of the total seed protein, while albumins approximately constitute 45% [23]. The antibacterial activity of 7S, 11S globulins and basic polypeptides from soybean have been recently investigated [24,25,26,27]. The general similarity between protein components of cowpea and those of other legumes may suggest similar functions and potential applications [28]. To further enrich the library of native antibacterial proteins with certain antimicrobial activities, the objective of the current research was to study the antibacterial activity of 7S and 11S globulins isolated from cowpea seed in vitro against Gram positive and Gram negative bacteria as compared to known specific antibiotics and in minced beef meat during preservation at 4 °C for 15 days.

## 2. Results

### 2.1. SDS-PAGE of 7S and 11S Globulins

The SDS-PAGE of cowpea native protein (CNP) as well as 7S and 11S globulin subunits of cowpea seeds are shown in Figure 1A. It is showed that CNP consists of two main fractions 7S and 11S globulins; the bands corresponding to 11S globulin (lane 3) are referring to about 28 and 52 kD_a_ indicating the basic and acidic subunits. The bands corresponding to 7S globulin (lane 2) are referring to about 132, 129 and 95 kDa, which indicate α^/^, α and β subunits.

### 2.2. Urea-PAGE of 7S and 11S Globulins

The migration in urea-PAGE in the cathode direction indicated that 11S and 7S globulins were much faster than their respective CNP, indicating bigger positive charges (Figure 1B).

### 2.3. pH-Solubility Curve of 7S and 11S Globulins:

The pH solubility curve of CNP, 7S and 11S globulins is given in Figure 1C. It is clear that the least soluble points were obtained at pH 4, 5 and 7 for CNP, 7S and 11S globulins, respectively.

### 2.4. Fourier Transform Infrared (FT-IR) Spectroscopy of 7S and 11S Globulins

The FT-IR spectra of CNP, 7S and 11S globulins from cowpea are shown in Figure 2. The secondary structure of the protein was commonly based on the amide I band analysis (1700–1600 cm^−1^). Amide I band peaks represented the most intense absorption bands of the polypeptides ν(C=O). There were also some in-plane NH bending contribution to amide I. The secondary structure of proteins is reflected by these bands as follows: 1610~1640 cm^−1^ for the β-sheet; 1640~1650 cm^−1^ for the random coil; 1650~1658cm^−1^ for the α-helix; 1660~1700 cm^−1^ for the β-turn.

### 2.5. Antibacterial Activity and MIC of 7S and 11S Globulins

Using disc diffusion method, it was demonstrated that 11S globulin had antibacterial activity against all the eight tested bacteria. The subunit 7S globulin demonstrated similar antibacterial activity against all the tested bacteria except for *E. coli*. The MICs values of 7S globulin (Table 1) against *S. typhi*, *K. pneumoniae*, *S. pyogenes*, *L. monocytogenes*, *L. ivanovii*, *P. aeruginosa and S. aureus* were 100, 200, 100, 10, 100, 50 and 200 µg mL^−1^ respectively; but they were of about 200, 400, 200, 200, 25, 25, 50 µg mL^−1^ respectively for 11S. Additionally, 11S was active against *E. coli* at 50 µg/mL.

### 2.6. Combined Effect of Cip-7S and Cip-11S against the Tested Bacteria

The antibacterial activity of cip: 7S and cip: 11S (at different ratios) showed increased antibacterial activity against *L. monocytogenes* and *S. aureus* respectively with increasing the proportion of protein (7S or 11S) but no evident synergistic effect between the two components can be seen (Table 2). The higher the protein concentration, the higher the inhibition zone was observed. There was a higher inhibition with 100 ug/mL of protein alone than with 100 μg/mL of ciprofloxacin alone. The diameter is lower when a lower concentration of proteins were used in combination with antibiotic.

### 2.7. TEM Image Analysis

TEM images of protein-treated bacteria given in Figure 3 show various signs of cellular deformation, indicating on direct disruptive action of 11S globulin on the cell wall and cell membrane. *P. aeruginosa* intact cells treated with 25 µg/mL of 11S globulin showed evidently reduced relative contents after 4 h of incubation at 37 °C and high mortality rates. The bacteria escaping the death were characterized by different manifestations of deformation, such as cell shrinkage, cell membrane wrinkles and pore formation as well as some emptiness of cellular live materials. *S. typhimurium* bacterial cells treated with 200 µg/mL of 11S globulin showed increased rate of damaged cells after their incubation at 37 °C for 4 h. The analysis of TEM images indicated that the cowpea seed protein globulins caused total degeneration of cell membranes, cell swelling, vacuole formation and finally complete lysis of cell components.

### 2.8. Storage of Minced Beef with 11S Globulin from Cowpea at 4 °C

#### Microbial Analysis

Minced meat samples were inoculated with 50 and 100 µg/g of 11S globulin and were stored at 4 °C for 15 days. Bacterial counts (CFU/mL) of total viable cells (TVC), coliform bacteria and psychotrophs were monitored throughout this storage period at time intervals of about 3 days. Results are given in Figure 4. Bacterial counts (CFU/g) of untreated meat samples (controls) increasesd and recorded ≥10^7^ CFU/g (≥7 log CFU/g) after 15 days of storage. However distinctive reduction in colony counts were detected from meat samples treated with 11S globulins (50 and 100 µg/g) after 3–15 days of storage at 4 °C. Difference in colony counts between untreated samples (controls) and treated samples were almost 3.6; 4.2 log CFU/g for TVC, and almost 4.8 log CFU/g; 6.9 log CFU/g for coliforms, and 4.5 log CFU/g; 4.5 log CFU/g for pyschrotrophs in meat samples treated with 50 and 100 µg/g respectively after 3 days of storage at 4 °C.

### 2.9. Physicochemical Analysis of Meat Sample

The untreated minced meat showed much increase of pH values than minced beef supplemented with 11S globulin (50 and 100 µg/g) after 15 days of storage at 4 °C as compared to zero time (Table 3). 

Minced meat supplemented with 11S globulin (50 and 100 µg/g) showed much reduced metmyoglobin percentage after 15 days of storage at 4 °C compared to zero or non-supplemented samples (Table 3). The rate of lipid oxidation inhibition was higher in meat samples supplemented with 11S globulin (50 and 100 µg/g) after 15 days of storage at 4 °C compared to zero time or control (Table 3).

## 3. Discussion

The high incidence of resistant bacterial variants to antimicrobial food additives has a vast impact on human mortality and healthcare [29]. Thus there is an urgent demand to find alternative antimicrobial food additives such as AMPs which are important components of innate immune systems. Antimicrobial peptides are highly active against most microbes, including both Gram-positive and Gram-negative bacteria [30,31]. Consequently, it is supposed that the AMPs are less bacterial resistance-generating than other antimicrobials [32].

The antimicrobial properties of plant proteins support their use as alternative food preservatives [33]. Several classes of plant proteins with antibacterial and/or antifungal properties have been isolated, identified and recommended as antimicrobial agents [8,9,24,34,35]. Therefore, there is a need to continue research to develop such safe antimicrobial proteins to be used as food preservatives with nutritive value. In this regard the 7S and 11S globulins of cowpea seeds were studied herein.

The appearance of two 7S globulin bands of about 28 and 52 kDa corresponds to the basic and acidic subunits and such results concur with other published work in this respect. Also, the molecular mass study of 7S globulin showed bands of about 132, 129 and 95 kDa and those bands correspond to α’, α and β subunits and those showed that such globulin protein is similar to that of chickpea [36]; SDS-PAGE of chickpea legumin exhibited five bands corresponding to smaller molecular weights of about 25–45 kDa in accordance with previous results in this respect which reported a molecular weight range of about 18-41 kDa for these subunits. The variations in the minimal molecular weight may be due to different plant varieties [31].

The faster migration of 11S and 7S globulins on urea-PAGE in the cathode direction than their respective CNP may indicate bigger positive charges. However, 7S globulin indicated relatively lower migration rate than 11S globulin in accordance with the fact that 11S globulin is originally more basic than 7S globulin [9,18]. These results are similar to urea-PAGE of chickpea proteins which indicated similar anode to cathode migration for chickpea legumin. Also, soybean glycinin protein showed fast migration to cathode, referring to similar high positive charge and basicity of chickpea legumin and soy glycinin [37].

The pH solubility curve of CNP, 7S and 11S globulins isolated from cowpea seeds showed the least soluble point at pH 4, 5 and 7 respectively. The relatively higher values of isoelectric points of 7S and 11S fractions (pH 5 and 7) than CNP (pH 4.5) might reflect their more basic nature, especially the second fraction (11S). Previous studies have showed that the isoelectric points of 7S and 11S globulins isolated from soybean was of about pH 5 and 6.5, respectively [15]. The similarity between protein components of soybean and those of other legumes suggests similar functions [28]. The secondary structure of CNP, 7S and 11S globulins was approved by FT-IR spectroscopy. This is coupled with the recent work about elucidation of the secondary structure of protein by FT-IR and to provide conformational and structure dynamic information of proteins [38].

The 11S globulin exhibited the largest IZD against all the tested bacteria. Also 7S globulin exhibited large IZD against all tested bacteria. The MICs values of 7S globulin (Table 1) against *S. typhi*, *K. pneumoniae*, *S. pyogenes*, *L. monocytogenes*, *L. ivanovii*, *P. aeruginosa and S. aureus* were 100, 200, 100, 10, 100, 50 and 200 µg mL^−1^ respectively; but they were of about 200, 400, 200, 200, 25, 25, 50 µg mL^−1^, respectively, for 11S. Additionally, 11S was active against *E. coli* at 50 µg/mL.

These results are similar to those obtained by previous studies [24]. The antimicrobial activity of CNP, 7S and 11S globulins of cowpea seeds could be due to their high positive charges and in turn hydrophobicity of such high molecular mass compounds which allows electrostatic interactions with the bacterial cellular components that affect the cells’ integrity. Consequently the bacterial cells lose their ability to divide and produce denergized killed cells [9,10].

The MIC data of both 7S and 11S globulins indicated almost some similar antibacterial effectiveness as that of methylated egg white proteins against some pathogenic bacteria [7]. Also, methylated soybean protein and methylated chickpea protein showed almost similar activities against *L. monocytogenes* and *S. enteritidis* [14]. Scientifically, comparison of MIC values of different compounds against different indicator bacteria could give misleading conclusions. Such a comparison could be made when authors of different research groups used similar experimental conditions and the same indicator organisms. The MIC values obtained of both 7S and 11S globulins (10–100 µg/g) against different pathogenic bacteria used could be promising in safe food preservation as food additives.

The high antibacterial activity of the protein components (7S and 11S) and cip. may provide grounds to formulate antibacterial drugs with certain proportions of the protein replacing the antibiotic. This may open the door for new therapeutic strategies involving less use of synthetic antibiotics in accordance with [24]. This result is promising for preparing partially or totally active protein-substituted antibiotic formations. The synergistic effect obtained between cowpea globulins and cip. could kill the resistant variants of pathogenic bacteria and could kill cells that escape the effect of each of them in separate use [7,9,10].

The analysis of TEM images indicated that the cowpea 11S globulins induced total degeneration of cell membranes, cell swelling, and vacuole formation leading finally to complete cell lysis. These results align with the previous reports of the direct action of cationic AMPs with cell membranes. This interaction occurs through electrostatic binding between the positively charged parts of the cationic proteins and the negatively charged layers of the cell membrane arising from teichoic acid and phospholipid components, triggering cell lysis. In Gram-negative bacteria this interaction is followed by insertion of the peptides into the outer membrane structure stimulated by hydrophobic interactions. The antibacterial peptides then cause destruction and permeabilisation of the cytoplasmic membrane [39,40].

Storage of minced meat with 11S globulin for 15 days at 4 °C reduced the growth of total viable count, psychrotrophic bacteria and coliforms as compared to controls. This is a promising result supporting the use of cowpea 11S globulin as a meat additive for processed meat products.

The preservative action of 11S globulin could be similar to that reported for soybean glycinin [18] and the chickpea legumin [37] due to its high antibacterial action against foodborne pathogens. The chemical structure of 11S globulin indicates its separation into basic and acidic domains, and might allow either domain to interact electrostatically with bacterial cellular components, affecting their cellular integrity [6,41]. The storage of control minced meat increased both the pH value and the metmyoglobin level probably due to the increased bacterial activity [42]. Adding globulins to stored minced meat limited these changes understandably as a result of the low bacterial activities achieved by adding these protein fractions [19,37]. Further work will be necessary to study the mode of action of both 7S and 11S globulins of cowpea in details and to study the impacts of their modification chemically for their preservative effect for meat and other foods.

## 4. Materials and Methods

### 4.1. Plant Material

*Vigna unguiculata* L.Walp. (Cowpea) seeds were purchased from a local market in Zagazig, Sharkia Governorate, (80 km north of Cairo, Egypt).

### 4.2. Microorganisms

Gram positive bacteria such as *L. monocytogenes* LMG 10470, *L. ivanovii* FLB 12, *S. aureus* ATCC 25923 and *S. pyogenes* ATCC 19615 and Gram negative bacteria such as *K. pneumoniae* ATCC 43816, *P. aeruginosa* ATCC 26853, *E. coli* ATCC 25922 and *S. typhimurium* ATCC 14028 were provided by the Laboratory of Applied Bacteriology, Department of Botany and Microbiology, Faculty of Science, Zagazig University, Egypt. All the indicator bacteria were stored in glass beads at –20 °C and were sub-cultured in brain heart infusion broth [25,26].

### 4.3. Isolation and Purification of Globulins from Cowpea (Vigna unguiculata) Seed

Cowpea seeds (1 kg) were ground and the resulting powder was defatted using *n*-hexane (5% *w/v*) for 8 h. The solvent was evaporated on a rotary-evaporator and dried-defatted meal was stored at 4 °C until analysis carried out. Total CNP was separated from defatted powder using the procedures described previously [23,27]. The 7S and 11S globulin protein subunits were isolated from the defatted powder of cowpea seeds according to the methods described previously with slight modifications [28]. Ten grams from the defatted cowpea seed meal were dispersed in 150 mL of buffer (0.03 mol/L tris HCl at pH 8.5, 0.4M NaCl, 10 mM β-mercaptoethanol, 1mM EDTA, 0.02% (*w/v*) NaN_3_) and the solubilized contents were separated by centrifugation at 5000 rpm for 10 min. The solution was stirred for 1 h at 45 °C in water bath. Then both 7S and 11S globulins were precipitated using ammonium sulphate (50–65% and 65–85%, respectively). The precipitate was dispersed and solubilized as described above and was dialyzed against the same buffer for 48 h to remove the salts.

#### 4.3.1. Fractionation of Papain Hydrolysed Cowpea

Papain hydrolysed cowpea (PHC) was fractionated by size exclusion chromatography (SEC) using a sephadex G-25 superfine grade resin (1.6 × 20 cm, Pharmacia™, Peapack, NJ, USA). A sample containing 40 mg mL^−1^ PHC was injected and eluted with potassium phosphate buffer (pH 7.0) at a flow rate of 5 mL min^−1^, and detected at 280 nm. Fractions of 10 mL were collected and lyophilised to evaluate their antibacterial activity.

#### 4.3.2. Fractionation of PHC by Size Exclusion Chromatography (SEC)

About 500 mg of the lyophilized PHC was dissolved in 15 mL of deionized water and separated for a second round on a Sephadex G-25 gel filtration column (1.6 × 150 cm). Two milliliters were injected and eluted with distilled water at a flow rate of 1 mL/min, and detected at 280 nm. The major peaks were collected and lyophilized to evaluate their antibacterial activity.

### 4.4. Characterization of 7S and 11S Globulins

#### 4.4.1. Sodium Dodecyl Sulfate Polyacrylamide Gel Electrophoresis (SDS-PAGE)

Twenty milligrams of CNP, 7S and 11S globulins from cowpea seeds were dissolved in 1 mL aliquots of SDS (10%) with 100 μL β-mercaptoethanol and subjected to intermittent vortexing during 15 min. The mixture was then centrifuged at 10,000× *g* for 5 min to separate the extract. Twenty μL of extract was mixed with 20 μL of SDS-loading sample buffer (SDS 4%, β-mercaptoethanol 3%, glycerol 20%, TrisHCl 50mM pH 6.8 and bromophenol blue traces), heated at 96 °C for 5 min and 10 μL aliquots were electrophoresed (10 μL of protein/ lane) and analyzed by SDS-PAGE [43].

#### 4.4.2. Urea-PAGE

Lyophilized CNP, 7S and 11S globulins from cowpea seeds were dispersed (10 mg/mL) in pH 6.8 buffer containing 0.25 M Tris-base, 50% glycerol and bromophenol blue traces. Samples were centrifuged at 15,000× *g* for 5 min at 20 °C. Supernatants were analyzed by urea-PAGE (10 μL of protein/lane) in 3% and 12% stacking and resolving gels, respectively [44].

#### 4.4.3. pH-Solubility Curve

Protein pH-solubility curves were assayed for CNP, 7S and 11S globulins from cowpea seeds in the pH 2–10 range [45]. The isoelectric points were estimated from the protein pH-solubility curves, as the pH corresponding to the least protein solubility [17,18].

#### 4.4.4. Fourier Transform Infrared (FT-IR) Spectroscopy

Protein samples were prepared with potassium bromide (KBr) pellet method [46]. Infrared spectra were measured with a FT-IR spectrometer (Nicolet Nexus 470, DTGS, Thermo Scientific, Waltham, MS, USA) at 25 °C. For each spectrum 256 interferograms were collected with a resolution of 4 cm^−1^ with 64 scans and a 2 cm^−1^ interval from the 4000 to 400 cm^−1^ region. The system was continuously purged with dry air. Second derivation spectra were obtained with Savitsky–Golay derivative function soft [47]. The relative amounts of different secondary structure of CPI, 11S and 7S globulins were determined from the infrared second derivative amide spectra by manually computing the areas under the bands assigned to a particular substructure.

### 4.5. Antibacterial Activity of 7S and 11S Globulins

The antibacterial activity of 7S and 11S globulins were determined against Gram positive bacteria (*L. monocytogenes*, *L. ivanovii*, *S. aureus* and *S. pyogenes*) and Gram negative bacteria (*K. pneumoniae*, *P. aeruginosa*, *E. coli* and *S. typhi*) by Kirby-Bauer disc-diffusion method [48]. Turbidity (at 600 nm) of the bacterial suspension was immediately measured after stirring the tube containing the bacterial suspension. The indicator bacteria were swabbed onto surface of brain heart infusion agar plates. Then, filter paper discs were soaked in globulins at different concentrations (10, 25, 50, 100, 200, 400, 800 and 1000 µg/mL) for 15 min and placed onto brain heart infusion agar (BHI, Oxoid, Basingstoke, UK) plates previously seeded with the indicator bacteria. After incubation for 24 h, at 37 °C, to reach about 10^6^ CFU mL^−1^. The diameter of inhibition zones (IZD) were measured using mm ruler and recorded after subtracting the diameter of the filter paper disc.

### 4.6. Determination of MIC Values of both 7S and 11S Globulins:

MIC values of both 7S and 11S globulins were determined using Kirby-Bauer disk-diffusion method [48]. The bacteria were swabbed on the surface of brain heart infusion agar plates. Then filter paper disks of 6-mm in diameter were soaked in diluted 7S solutions (10, 25, 50, 100, 200, 400, 800 and 1000 μg/mL) and 11S solution (10, 25, 50, 100, 200, 400, 800 and 1000 μg/mL) and were then placed onto agar surface with suitable distances separating them from each. The plates were incubated at 37 °C for 24 h and IZD were measured using a millimeter ruler. MIC was defined as the least concentration inducing bacterial inhibition.

### 4.7. Antibacterial Activity of Combinations between the Ciprofloxacin Antibiotic and the Globulins

In the current study, ciprofloxacin (cip) (750 mg, E.I.P.I.Co, 10th of Ramadan City, Egypt) was selected as a high antibacterial broad-spectrum antibiotic and *L. monocytogenes* and *S. aureus* were selected as highly sensitive strains to 7S and 11S globulins. Using standard disc diffusion method, the cip antibiotic (100 µg/mL) was mixed with either 7S or 11S globulins (100 µg/mL) to determine the antibacterial activity of the different combinations against *L. monocytogenes* and *S. aureus* to finally conclude any potential synergistic effect or even natural substituting possibility. The cip: globulin combinations were prepared at varying percentages; (i.e., 100% cip + 0% globulins; 80% cip + 20 % globulins; 60% cip + 40% globulins; 40 cip + 60% globulins; 20 cip + 80% globulins; and 0.0 % cip + 100% globulins) of its starting concentration. The antibacterial activity was determined as previously demonstrated [7,18].

### 4.8. Transmission Electron Microscopy (TEM)

*S. typhimurium* and *P. aeruginosa*, were selected for TEM examination because the mode of action of similar globulins was studied against other pathogenic bacteria previously by the same research group [17,18]. These bacteria were grown in brain heart infusion broth (Oxoid) incubated at 37 °C to reach about 10^6^ CFU mL^−1^. The MICs values of about 25 μg/mL and 200 μg/mL of 11S globulin were added to *S. typhimurium* and *P. aeruginosa* cell suspensions respectively except controls (bacteria without globulin treatment) and incubated at 37 °C for 4 h. Ultra thin sections were prepared for investigation by TEM. Perfusion or immersion fixation of the tissue occurred using a previously method [49]. The cells were left overnight at 4 °C, then washed 3× for 15 min in 0.1 M sodium phosphate buffer + 0.1 M sucrose and post-fixed 90 min. in 2% sodium phosphate buffered osmium tetroxide pH 7.4. Then washed 3× for 15 min in 0.1 M sodium phosphate buffer pH 7.4 and dehydrate 2× 15 min: 50% ethanol (in distilled water). Then contrasted overnight using 70% acetone + 0.5% uranyl acetate + 1% phosphotungstic acid at 4 °C, 2× for 15 min. 80 % ethanol, 2× 15 min. 90 % ethanol, 2× for 15 min. 96 % ethanol, 3 × 20 min. 100% ethanol and 2 × 15 min. acetone. Then 30 min. 2:1 acetone:Epon mixture, 30 min. 1:1 acetone:Epon mixture, 30 min. 1:2 acetone:Epon mixture, Epon pure solution overnight at 4 °C and finally new fresh Epon solution. After that they were put in incubator for 48 h. at 65 °C for polymerization and cut with an ultra microtome set to 50–100 nm section thickness. Then rinse sections to grids or gelatine-covered one-whole grids made of cooper or nickel.

Post contrasting of sections were carried out as reported previously: 10 min. 8% uranyl acetate and 5 min 0.7% leadcitrate + 0.9% sodium citrate after drying for 15 min sections may be investigated in a transmission electron microscope [50]. Ultrathin sections were observed at 80 kV using a JEOL 2100 TEM at 80 KV at EM Unit, Mansoura University, Mansoura, Egypt.

### 4.9. Storage of Minced Beef with 11S Globulin Isolated from Cowpea at 4 °C

#### 4.9.1. Meat Samples

The fresh raw beef purchased from local market in Zagazig city, Sharkia Governate, Egypt was minced in sanitized meat mincers (Moulinex Co., Cairo, Egypt). The samples of the minced meat were subjected to further work.

#### 4.9.2. Experimental Design

Minced beef samples (100 g) were placed in sterilized polyethylene bags (Gomhoria Co., Zagazig, Egypt) and were homogenized using stomacher (New Brunswick Scientific Co., Edison, NJ 08818-4005, USA.) for 2 min at room temperature. The 11S globulin (50 and 100 µg/g) was mixed with minced meat and the treated samples were subjected to homogenization for 2 min. using the same stomacher. Untreated sample from each treatment was used as control. Containers with samples from all treatments and controls were stored under aerobic conditions at 4 °C for 15 days. Microbial analysis and physicochemical analysis were carried out at different intervals of preservation (0, 3, 6, 9, 12 and 15 days) at 4 °C.

#### 4.9.3. Microbial Analysis

Bacterial counts (CFU/mL) were monitored from both samples and controls each 3 days of preservation at 4 °C through the storage period (0–15 days) following the procedures outlined previously [51,52]. About 10 g portions of both treated and untreated meat samples were withdrawn aseptically and transferred to a sterile flasks, each containing 90 mL of sterile peptone (Bhoidapada, Vasai East Mumbai, India) water (1.0 g peptone + 8.5 g NaCl in 1 L of distilled water) and were then shaken vigorously by hand for homogenization. From this 1:10 dilution, serial two-fold dilutions were made. Colony forming units (CFU/mL) were determined for viable bacteria; coliforms; psychrotrophs after incubation onto nutrient agar (Oxoid); MacConkey agar; nutrient agar (Oxoid) at 25 °C for 72 h.; 37 °C for 24 h.; 7 °C for 10 days respectively.

### 4.10. Physicochemical Analysis of Meat Sample

#### 4.10.1. pH Determination

The pH values were determined for homogeneous mixtures of meat with distilled water (1:10, *w*/*v*) [53]. Five grams portions of the meat samples were homogenized in 50 mL distilled water and the mixture was filtered. Then pH of the filtrate was measured using pH-meter (pH 211 Hansen Instruments Inc., Woonsocket, RI, USA) at each sampling point.

#### 4.10.2. Metmyoglobin (MetMb) Analysis

The contents of MetMb in the ground beef samples were determined as described previously [54]. Briefly, 5 g samples were homogenized in 25 mL ice-cold 40 mM phosphate buffer (pH 6.8) for 10 s. The homogenate was kept for 1 h at 4 °C and centrifuged at 4500 g for 30 min at 4 °C. The supernatant was filtered through Whatman filter paper No. 1 and the absorbance was measured at 572, 565, 545 and 525 nm using a 6405 UV/visible spectrophotometer (Jenway, Staffordshire, ST15 OSA, UK). Then the percentages of MetMb were calculated based on these absorbance values according to [55] using the following formula:
(1)
MetMb (%)=[−2.51(A572A525)+0.777(A565A525)+0.8(A545A525)+1.098]×100


#### 4.10.3. Lipid Peroxidation Assay

Lipid peroxidation in minced beef supplemented with 11S globulin at both concentrations (50 and 100 µg/g) was measured after different intervals of preservation (0–15 days) at 4 °C [56]. Five grams of each meat sample were homogenized and a volume of 10 % *w*/*v* homogenate was prepared in 0.05 M phosphate buffer (pH 7) and centrifuged at 12,000× *g* for 60 min at 4 °C. The supernatant obtained was used for lipid peroxidation assessment. An aliquot of 100 µL from supernatant was treated with 2000 µL of (1:1:1 ratio) TBA–TCA–HCl reagent (thiobarbituric acid 0.37%, 15% trichloroacetic acid and 0.25 N HCl). All the tubes were placed in a boiling water bath for 30 min and allowed to cool. The optical density of the supernatant was measured at 535 nm using Jenway 6405 UV/visible spectrophotometer against a reagent blank. Percentage inhibition was calculated using the following equation:
(2)
Lipid oxidation inhibition (%)=[1−(absorbance of sample/absorbance of control)]×100


## 5. Conclusions

Based on the obtained results, it can be concluded that cowpea globulins are effective antibacterial agents that can be successfully and efficiently used as a meat preservative, particularly as they are safe natural products and can be prepared at considerably low costs, apart from their known rich nutritional character as legume proteins.

## Figures and Tables

**Figure 1 molecules-24-01082-f001:** Biochemical characters of cowpea proteins. (**A**) SDS-PAGE; where St = Standard protein, Lane 1: cowpea native protein (CNP), Lane 2: 7S globulin, Lane 3: 11S globulin, AS: Acidic subunit, BS: basic subunit. (**B**) UREA-PAGE and (**C**) pH solubility of CNP.

**Figure 2 molecules-24-01082-f002:** FT-IR different spectra of cowpea native protein (CNP), and its subunits; 11S and 7S globulins.

**Figure 3 molecules-24-01082-f003:** TEM of *P. aeruginosa* after their incubation at 37 °C for 4 h affected by 25 µg/mL of 11S globulins and *S. typhimurium* after their incubation at 37 °C for 4 h affected by 200 µg/mL of 11S globulins.

**Figure 4 molecules-24-01082-f004:** Total viable count, psychotrophs count and coliform count in minced beef treated with 11S globulin at different concentrations (50 and 100 µg/g) and stored at 4 °C for different periods (0–15 day) compared to the control.

**Table 1 molecules-24-01082-t001:** Antibacterial activity of 7S and 11S globulins at different concentrations against different pathogenic bacteria.

Bacterial Strains	Protein Subunit Concentration (µg/mL)
10	25	50	100	200	400	800	1000
	Inhibition Zone Diameter (mm)
7S globulin
*S. typhimurium*	-	-	-	10 ± 0.58	10.3 ± 0.75	10.6 ± 0.57	13 ± 1.15	19 ± 0.15
*K. pneumoniae*	-	-	-	-	9 ± 0.57	10.3 ± 0.58	10.6 ± 0.57	11.6 ± 0.57
*S. pyogenes*	-	-	-	10.3 ± 0.75	11.3 ± 0.74	13.3 ± 0.75	11.6 ± 0.75	13.6 ± 0.92
*L.* *monocytogenes*	13.6 ± 0.35	15 ± 1.15	15 ± 1.15	16 ± 1.15	19.3 ± 1.15	20 ± 1.15	20 ± 1.14	23 ± 1.73
*L. ivanovii*	-	-	-	9 ± 0.58	10.6 ± 0.92	11 ± 0.57	13 ± 0.88	18 ± 1.15
*P. aeruginosa*	-	-	10.3 ± 0.58	11 ± 0.86	12.6 ± 1.50	17 ± 1.15	18 ± 0.86	19 ± 0.86
*S. aureus*	-	-	-	0.0 ± 0.0	9 ± 0.56	10.6 ± 0.58	14.6 ± 0.57	15 ± 0.58
*E. coli*	-	-	-	-	-	-	-	-
11S globulin
*S. typhi*	-	-	-	-	12.6 ± 0.92	13.6 ± 0.92	13 ± 1.15	23 ± 1.45
*k. pneumonia*	-	-	-	-	0.0 ± 0.0	9.3 ± 0.58	9.6 ± 0.57	14 ± 0.86
*St. pyogenes*	-	-	-	-	8 ± 0.57	10 ± 0.57	11 ± 0.86	13.6 ± 0.58
*L. monocytogens*	-	-	-	-	7.6 ± 0.35	9.3 ± 0.56	9.6 ± 0.57	13.6 ± 0.58
*L. ivanovii*	-	8.7 ± 0.40	11 ± 0.86	12.3 ± 0.75	15.3 ± 0.75	23 ± 1.73	19 ± 1.15	25 ± 1.73
*P. auriginosa*	-	9 ± 0.57	10.3 ± 0.58	11 ± 0.57	11.6 ± 0.58	13 ± 0.86	13.3 ± 0.75	15 ± 0.86
*S. aureus*	-	-	14.6 ± 1.15	15.3 ± 1.15	0.0 ± 0.0	24 ± 1.15	24 ± 1.15	25 ± 1.73
*E. coli*	-	-	9 ± 0.57	9.3 ± 0.57	10 ± 0.57	10.3 ± 0.58	10 ± 0.57	17.3 ± 1.15

**Table 2 molecules-24-01082-t002:** Antibacterial activity against *L. monocytogenes* and *S. aureus* using combinations of antibiotic (Cip.) and both of 7S and 11S globulin, as assessed by disc diffusion assay.

Concentration (µg/mL)	Inhibition Zone Diameter (mm)
7S	11S
Against
Cip	Protein	*L. monocytogenes*	*S. aureus*
100	0	15.3 ± 0.58	11.3 ± 0.57
80	20	17.0 ± 0.69	17.6 ± 0.93
60	40	17.3 ± 0.75	20.0 ± 1.15
40	60	19.4 ± 0.31	28.0 ± 1.15
20	80	20.8 ± 0.51	30.0 ± 1.44
0	100	24.5 ± 0.52	32.6 ± 1.50

**Table 3 molecules-24-01082-t003:** Physicochemical analysis of meat samples.

Storage Time (day)	Control	11S (µg/g)
50	100
pH Values
0	5.6	5.6	5.6
3	5.9	5.7	5.7
6	6.5	5.8	5.75
9	6.6	6.3	5.8
12	6.9	6.4	6.1
15	7.2	6.55	6.3
	Metmyoglobin (%)
0	11	11	11
3	34.6	24	20
6	50.6	30	25
9	55	39	30
12	63	45	32
15	69	50	45
	Lipid Oxidation Inhibition (%)
0	25	25	25
3	23	24	24
6	22	23	24
9	19	22	23
12	13	18	20
15	10	16	17

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
