# Peer review of "Characterization and Antibacterial Activity of 7S and 11S Globulins Isolated from Cowpea Seed Protein"

_molecules, 2019, doi:10.3390/molecules24061082_

Reviewer 1 Report

In this paper the authors investigated the antibacterial activity of 7S and 11S globulins isolated from cowpea seed. The introduction doesn't provide sufficient background and doesn't include all relevant references. In Results the Table 1 is not absolutely clear and therefore must be done again.

As regard Table 2, I would like to add "Antibacterial activity against L..."

As regard Figure 3, the figures must be better aligned.

In Methods, Section 4.3.2. Fractionation of PHC... and not CPH

In Methods, Section 4.7. "ciprofloaxacin" must be corrected in "ciprofloxacin"

In Methods, Section 4.9.3. Microbial analysis, "nutireint" must be corrected in "nutrient"

The Conclusion must be improved.

Author Response

Reviewer   # 1                                                                    

In this paper the   authors investigated the antibacterial activity of 7S and 11S globulins   isolated from cowpea seed. The introduction doesn't provide sufficient   background and doesn't include all relevant references.

The introduction was revised and   enriched.

In Results the Table 1   is not absolutely clear and therefore must be done again. 

Table   1 was reshaped

As regard Table 2, I   would like to add "Antibacterial activity against L..." 

(Antibacterial activity   against) was added.

As regard Figure   3, the figures must be better aligned.

Figure 3 was   realigned.

In Methods, Section   4.3.2. Fractionation of PHC... and not CPH

Done

In Methods, Section   4.7. "ciprofloaxacin" must be corrected in   "ciprofloxacin"

Done

In Methods, Section 4.9.3.   Microbial analysis, "nutireint" must be corrected in   "nutrient"

Done

The Conclusion must be   improved.

Done

Reviewer 2 Report

Manuscript ID: molecules-457505

Characterization and antibacterial activity of 7S and 11S globulins isolated from cowpea seed protein

In this work, Abdel-Shafi and co-authors studied two globulins isolated from cowpea seeds as potential antimicrobial food preservatives. For that, the authors have extracted, isolated, purified and chemically characterized the globulins from Vigna unguiculata (L.) Walp. seeds. The antibacterial activity and minimum inhibitory concentrations of 7S and 11S globulins was evaluated against  4 Gram-positive bacteria and 4 Gram-negative bacteria. A synergic effect of the globulins with a reference antibiotic was also tested. Furthermore, the effect of the antimicrobial treatment was evaluated by TEM to get an insight on the mechanism of action of the 11S globulin. Finally, the protective antibacterial effect of 11S globulin as food preservative was tested on minced beef for up to 15 days at 4ºC. The authors concluded that 7S and 11S globulins from cowpea seed protein have antibacterial activity and are promising potential meat preservatives.

This manuscript is well designed and reports a lot of experimental work. It has scientific merit and fits in the scope of Molecules. However, there are some major issues to be considered before being considered for publication, related to scientific soundness, quality of presentation and significance of content. English language should be revised, the presentation of the subject in the Introduction needs an ortho-logical sequence, the methods and results are poorly detailed, the discussion has to be rewritten, and the bibliographic references must be carefully chosen. Please see the comments below.

Specific comments:

Language and writing: English language, grammar, spelling, punctuation, formats, abbreviations, hyphenations, the use of parenthesis, need to be extensively and carefully revised. Besides, the language has to be more objective and clearer, it cannot be redundant.

Misuse of references: every reference should be carefully revised. Sometimes is seems they were placed randomly in the text. I can give a few examples:

1)    line 46: to say that cowpea is an especially important source of protein in the lower income segment of the population, a citation about the validity of raw buffalo milk by esterified legume proteins is used;

2)    line 50: to refer the composition of 11S globulin, the authors cite a reference about milk quality deterioration by mild thermization with methylated chickpea protein;

3)    line 53: for the recent antibacterial activity testing of 7S and 11S globulins and basic polypeptides, a reference from of 1985 is cited (ref. 26);

4)    line 148, ref 53: Is this reference supposed to represent a survey of a number of studies worldwide?

5)    line 160: reference 56 is about heavy metal contents and magnetic properties of street dust in Hong Kong;

6)    other misplaced references: line 142, ref 49; line 145, refs 50 and 51; and many others.

Abstract: there is no need to use abbreviations of the bacterial species in parenthesis. They are abbreviated after the first use written in full.

Line 26: “distinctive decrease”: what does this mean in terms of bactericidal effect? Was it bactericidal? If so, please state.

Introduction: The introduction has to be restructured. Authors should present ideas in a logical and connected way. In this case, ideas are presented without a guiding line, although they are correct. Only relevant references inserted in the text should be cited in an appropriate way. It seems to me there is an abuse of self-citations. This is not a review article, so excessive use of bibliographic references should be avoided. Authors should choose the most appropriate references to the subject they are developing. This happens at the end of the first paragraph, line 34; at the end of the second, line 37; and at the end of the third. In the latter case, the authors note the importance of using new food preservatives citing many references without specifying what these preservatives are and what has already been discovered. Missing a paragraph to justify this and the references used. Use hyphen for multiple references, for example, instead of 1,2,3,4,5, use 1-5.

“Methods” section

Avoid the use of citations here, unless strictly necessary. Instead, briefly describe your procedures. Otherwise, the readers won’t go search for all the references looking for the methods.

Line 232: Why to cite references 30 and 53 here? Instead, refer the brand, city and country of the manufacturer of the growing medium;

Line 234: “cowpea seeds had been ground” - How much of sample?

Line 234: “using n-hexane” – how? Which volume? Please detail de experimental conditions;

Line 236: After reference 31, please describe briefly the experimental;

Line 251 and 252: CPH and PHC – please uniformize these abbreviations and all others throughout the text;

Line 286: “indicator bacteria” – What concentration? Did you measure the optical density of the bacterial suspension?

Line 287: “filter paper disks were soaked in globulins” - What concentrations? Separately? Please detail.

Line 288: “after incubation for 24h” – At which temperature?

These details are somehow mentioned in the “Results” section but they should be here.

Line 290: A paper of electrophoresis (ref. 38) talked about the measurement of inhibition zone diameters? Again, there is no need to place a reference here. The same for line 297;

Line 301: “the former was the more effective antibiotic” – How? Towards what?

Line 303: For ciprofloxacin refer the concentrations used (in molar or mass/volume) and reference, brand, city and country of the manufacturer;

Line 304: the meaning of this sentence is not perceived;

Lines 305-307: refer the concentrations used;

Line 314: “except controls” – please explain in what the controls consisted of;

Line 316: “using a modified procedure (40).” – a procedure modified from ref. 40? Please, describe briefly;

Line 343: “different intervals of preservation (0-15 days) at 4ºC” – 1 per day? Which were the intervals? This should be clear here;

Line 347: Is reference 41 correct?

Line 347: Here you cite reference 42, the Kirby-Bauer diffusion disk method. Why didn´t you cite this in sections 4.5, 4.6 and 4.7?

Line 348: “peptone water” – reference, brand, city na dcountry of the manufacturer;

Line 353: Please remove the references;

Section 4.10.1: Please remove the references and “made in Romania”, not necessary.

Line 359: Which were the sampling points? Please detail.

“Results” section

Line 64: “11S globulin ‘(lane 3)’ ”

Line 65: “7S globulin ‘(lane 2)’ ”

Line 70: legend for AS and BS?

Line 62: CNP should be written in full and appear abbreviated for the first time here and not in line 235;

CNP and CPI are both used in the text. What is the difference?

Line 87: A section number is missing here. It would be better as “Antibacterial activity and MIC of 7S and 11S globulins”;

Limes 88-90: This belongs to the Discussion section;

Lines 97-99: Describe your results. This is Discussion and not exactly true. With ciprofloxacin, there was inhibition but the higher the protein concentration, the higher the inhibition, even decreasing the antibiotic concentration. There was a higher inhibition with 100 ug/mL of protein alone than with 100 ug/mL of ciprofloxacin alone. The diameter is lower when a lower concentration of proteins are used in combination with antibiotic.

Section 2.6) S. typhi or S. typhimurium? Not clear.

Line 110: “after their incubation at 37ºC for 4h” – this should be also on the legend of Fig. 3;

Line 113: What is P. aeruginosa as?

Line 116) This section is not numbered;

Line 117: “50 and 100 ug/g” – of what?

Line 122: “P value” – the authors performed a statistical analysis, so a sub-section describing the analysis should be included in the “Methods” section;

Line 124: 4.8 log CFU/g; 6.9 log CFU/g of coliforms” – I cannot see this decrease for the coliforms after 3 days. Maybe after 15 days. The same for psycotrophs. After 6 days and not after 3 days the difference stated can be observed. In this section, and from the 3rd day on? What information can be retrieved from these analyses from day 6, 9, 12, 15, and overall? Describe better your findings. You’ve done a lot of work.

Lines 134-138: Are the values significantly different from the control? A statistical analysis has been performed?

“Discussion” section:

Line 160: what is this “variant effect”?

There are many references used incorrectly. First, you should discuss your results and then compare with other studies. The text reports ideas from many different articles without making a logical connection or comparison with the results obtained in this work;

Line 176-177: mention the concentrations and discuss the dose effect;

Lines 178.182: This is speculative about the mechanism of action of globulins on bacteria and not a true discussion of the results. Which was the most susceptible strain to the treatment? What is the most effective concentration of antibiotic on Gram-positive and on Gram-negative bacteria?

Lines 192-193: This sentence has no grammatical sense;

Line 208: “reduced significantly” – where is the statistical analysis described?

Lines 209-211: This conclusion does not make sense the way it is.

“Conclusion” section:

Line 382 – “relatively safer” – than what? A comparison has been made?

Line 382 – “relatively low price” – Compared to what?

Lines 382-383: “rich nutritional value” – Was this evaluated? This is not a conclusion from this study.

“References” should be carefully revised in terms of spelling, spacing, journal abbreviations, italics, etc. There are many errors.

Author Response

Reviewer   # 2

In   this work, Abdel-Shafi and co-authors studied two globulins isolated from   cowpea seeds as potential antimicrobial food preservatives. For that, the   authors have extracted, isolated, purified and chemically characterized the   globulins from Vigna   unguiculata (L.) Walp. seeds.   The antibacterial activity and minimum inhibitory concentrations of 7S and   11S globulins was evaluated against  4 Gram-positive bacteria and 4   Gram-negative bacteria. A synergic effect of the globulins with a reference   antibiotic was also tested. Furthermore, the effect of the antimicrobial   treatment was evaluated by TEM to get an insight on the mechanism of action   of the 11S globulin. Finally, the protective antibacterial effect of 11S   globulin as food preservative was tested on minced beef for up to 15 days at   4ºC. The authors concluded that 7S and 11S globulins from cowpea seed protein   have antibacterial activity and are promising potential meat preservatives.

Ok

This   manuscript is well designed and reports a lot of experimental work. It has   scientific merit and fits in the scope of Molecules. However, there are some   major issues to be considered before being considered for publication,   related to scientific soundness, quality of presentation and significance of   content. English language should be revised, the presentation of the subject   in the Introduction needs an ortho-logical sequence, the methods and results   are poorly detailed, the discussion has to be rewritten, and the   bibliographic references must be carefully chosen. Please see the comments   below.

English language was revised

Orthological sequence was considered in the introduction.

Specific   comments:

Language and writing: English language, grammar, spelling,   punctuation, formats, abbreviations, hyphenations, the use of parenthesis,   need to be extensively and carefully revised. Besides, the language has to be   more objective and clearer, it cannot be redundant.

These comments were considered throughout the manuscript

Misuse of references: every reference should be carefully   revised. Sometimes is seems they were placed randomly in the text. I can give   a few examples:

References were revised in all the manuscript.

1)    line   46: to say that cowpea is an especially important source of protein in the   lower income segment of the population, a citation about the validity of raw   buffalo milk by esterified legume proteins is used;

References were   revised in all the manuscript.

2)    line   50: to refer the composition of 11S globulin, the authors cite a reference   about milk quality deterioration by mild thermization with methylated   chickpea protein;

References were   revised in all the manuscript.

3)    line   53: for the recent antibacterial activity testing of 7S and 11S globulins and   basic polypeptides, a reference from of 1985 is cited (ref. 26);

References were   revised in all the manuscript.

4)    line   148, ref 53: Is this reference supposed to represent a survey of a number of   studies worldwide?

References were   revised in all the manuscript.

5)    line   160: reference 56 is about heavy metal contents and magnetic properties of   street dust in Hong Kong;

References were   revised in all the manuscript.

6)    other   misplaced references: line 142, ref 49; line 145, refs 50 and 51; and many   others.

References were   revised in all the manuscript.

Abstract:   there is no need to use abbreviations of the bacterial species in   parenthesis. They are abbreviated after the first use written in full.

This was revised and done accordingly.

Line 26: “distinctive decrease”: what does this mean in terms of   bactericidal effect? Was it bactericidal? If so, please state.

In this context the antibacterial effect was just evident.

Introduction:   The introduction has to be restructured. Authors should present ideas in a   logical and connected way. In this case, ideas are presented without a   guiding line, although they are correct. Only relevant references inserted in   the text should be cited in an appropriate way. It seems to me there is   an abuse of self-citations. This is not a review article,   so excessive use of bibliographic references should be avoided. Authors   should choose the most appropriate references to the subject they are   developing. This happens at the end of the first paragraph, line 34; at the   end of the second, line 37; and at the end of the third. In the latter case,   the authors note the importance of using new food preservatives citing many   references without specifying what these preservatives are and what has   already been discovered. Missing a paragraph to justify this and the   references used. Use hyphen for multiple references, for example, instead of   1,2,3,4,5, use 1-5.

1,2,3,4,5,   use 1-5.= done.      Reference were checked. Self-citation   is used when is relevant and serving the subject.

“Methods” section

Avoid   the use of citations here, unless strictly necessary. Instead, briefly   describe your procedures. Otherwise, the readers won’t go search for all the references   looking for the methods.

Done

Line   232: Why to cite references 30 and 53 here? Instead, refer the brand, city   and country of the manufacturer of the growing medium;

References were   revised in all the manuscript.

Line   234: “cowpea seeds had been ground” - How much of sample?

Done- Cowpea seeds (1 kg) had been ground and   the resulting powder was defatted using n-hexane (5% w/v) for 8 h. Solvent was evaporated by   rotary-evaporator and dried-defatted meal was stored at 4 °C until analysis   carried out. Total   CNP

Line   234: “using n-hexane” – how? Which volume? Please detail de experimental   conditions;

Done Cowpea seeds (1 kg) had been ground and the resulting powder   was defatted using n-hexane (5%   w/v) for 8 h. Solvent was evaporated by rotary-evaporator and dried-defatted   meal was stored at 4 °C until analysis carried out. Total CNP

Line   236: After reference 31, please describe briefly the experimental;

Done

Line   251 and 252: CPH and PHC – please uniformize these abbreviations and all   others throughout the text;

Done

Line   286: “indicator bacteria” – What concentration? Did you measure the optical   density of the bacterial suspension?

Yes, turbidity   (at 600 nm) of the bacterial suspension is immediately measured after   stirring the tube containing the bacterial suspension.

Line   287: “filter paper disks were soaked in globulins” - What concentrations?   Separately? Please detail.

Done- at different concentrations (10, 25, 50, 100, 200, 400,   800 and 1000 µg/mL)

Line   288: “after incubation for 24h” – At which temperature?

at 37 ºC,

These   details are somehow mentioned in the “Results” section but they should be   here.

Line   290: A paper of electrophoresis (ref. 38) talked about the measurement of   inhibition zone diameters? Again, there is no need to place a reference here.   The same for line 297;

Done

Line   301: “the former was the more effective antibiotic” – How? Towards what?

This part was reformulated for better clarity.

Line   303: For ciprofloxacin refer the concentrations used (in molar or   mass/volume) and reference, brand, city and country of the manufacturer;

Done  

Line   304: the meaning of this sentence is not perceived;

This sentence was reformulated.

Lines   305-307: refer the concentration used;

Done-(750 mg, E.I.P.I.Co, 10th of Ramadan City,   Egypt) was selected as a high antibacterial broad-spectrum antibiotic and L. monocytogenes and S. aureus were   selected as highly sensitive strains to 7S and 11S globulins. Using standard disc diffusion   method, the cip antibiotic (100 µg/mL) was   mixed with either 7S or 11S globulins (100 µg/mL) to determine the antibacterial activity of the   different combinations against L. monocytogenes and S. aureus   to finally conclude any potential synergistic effect or even natural   substituting possibility.

Line   314: “except controls” – please explain in what the controls consisted of;

Controls mean bacteria without treatment by   globulins

Line   316: “using a modified procedure (40).” – a procedure modified from ref. 40?   Please, describe briefly;

using a previously method

Line   343: “different intervals of preservation (0-15 days) at 4ºC” – 1 per day?   Which were the intervals? This should be clear here;

Done- Microbial analysis and physicochemical analysis were carried out at   different intervals of preservation (0, 3, 6, 9, 12 and 15 days) at 4 °C. 

Line   347: Is reference 41 correct?

References were   revised in all the manuscript.

Line   347: Here you cite reference 42, the Kirby-Bauer diffusion disk method. Why   didn´t you cite this in sections 4.5, 4.6 and 4.7?

References were   revised in all the manuscript.

Line   348: “peptone water” – reference, brand, city na dcountry of the   manufacturer;

Done

Line   353: Please remove the references;

Done

Section   4.10.1: Please remove the references and “made in Romania”, not necessary.

Line   359: Which were the   sampling points? Please detail.    دعلى

Done

“Results” section

Line   64: “11S globulin ‘(lane 3)’ ”

Line   65: “7S globulin ‘(lane 2)’ ”

Modified accordingly

Line   70: legend for AS and BS?

AS: Acidic subunit, BS basic subunit.

Added.

Line 62: CNP should be written in full and appear abbreviated   for the first time here and not in line 235;

Done.

CNP and   CPI are both used in the text. What is the difference?

They are referring to the same compound. So, CPI was replaced by CNP   all over the text.

Line   87: A section number is missing here. It would be better as “Antibacterial   activity and MIC of 7S and 11S globulins”;

Done accordingly

Limes   88-90: This belongs to the Discussion section;

Lines 88-90 (Using disc diffusion method,   it was demonstrated that 11S globulin had antibacterial activity against all   the eight tested bacteria. The subunit 7S globulin demonstrated similar   antibacterial activity against all the tested bacteria except for E.   coli.) seem rather results. Its removal from this place will affect   understanding the results.

Lines   97-99: Describe your results. This is Discussion and not exactly true. With   ciprofloxacin, there was inhibition but the higher the protein concentration,   the higher the inhibition, even decreasing the antibiotic concentration.   There was a higher inhibition with 100 ug/mL of protein alone than with 100   ug/mL of ciprofloxacin alone. The diameter is lower when a lower   concentration of proteins are used in combination with antibiotic.

Yes, you are right. The increased activity of the combination   may be due to the increased proportion of the protein and not to a   synergistic effect. So this part was modified as follows (The antibacterial activity of   cip: 7S and cip: 11S (at different ratios) showed increased antibacterial   activity against L. monocytogens and S. aureus respectively   with increasing the proportion of protein (7S or 11S) but no evident   synergistic effect between the two components can be seen (Table 2).   The higher the protein concentration, the higher the inhibition zone was   observed.  There was a higher   inhibition with 100 ug/mL of protein alone than with 100 ug/mL of   ciprofloxacin alone. The diameter is lower when a lower concentration of   proteins are used in combination with antibiotic.)

Section   2.6) S. typhi or S. typhimurium? Not clear.

It is S. typhimurium. Corrected all over the text

Line   110: “after their incubation at 37ºC for 4h” – this should be also on the   legend of Fig. 3;

Line   113: What is P. aeruginosa as?

Line   116) This section is not numbered;

Line   117: “50 and 100 ug/g” – of what? of 11S globulin

Done

Line   122: “P value” – the authors performed a statistical analysis, so a   sub-section describing the analysis should be included in the “Methods”   section;

We delete (P.   value ˂ 0.05). we don't make significant   analysis.

Line   124: 4.8 log CFU/g; 6.9 log CFU/g of coliforms” – I cannot see this decrease   for the coliforms after 3 days. Maybe after 15 days. The same for   psycotrophs. After 6 days and not after 3 days the difference stated can be   observed. In this section, and from the 3rd day on? What   information can be retrieved from these analyses from day 6, 9, 12, 15, and   overall? Describe better your findings. You’ve done a lot of work.

Lines   134-138: Are the values significantly different from the control? A   statistical analysis has been performe

we don't make significant analysis.

“Discussion” section:

 Line   160: what is this “variant effect”?

to   different plant varieties

There   are many references used incorrectly. First, you should discuss your results   and then compare with other studies. The text reports ideas from many   different articles without making a logical connection or comparison with the   results obtained in this work;

References were revised in all   the manuscript.

Line   176-177: mention the concentrations and discuss the dose effect;

Done- The MICs values of 7S globulin (Table 1) against S.   typhi, K .pneumoniae, S. pyogenes, L. monocytogenes, L. ivanovii, P. aeruginosa   and S. aureus were 100, 200, 100, 10, 100, 50 and 200 µg mL-1   respectively; but they were of about 200, 400, 200, 200, 25, 25, 50 µg mL-1  respectively for 11S. Additionally, 11S was   active against E. coli at 50 µg/mL.

Lines   178.182: This is speculative about the mechanism of action of globulins on   bacteria and not a true discussion of the results. Which was the most   susceptible strain to the treatment? What is the most effective concentration   of antibiotic on Gram-positive and on Gram-negative bacteria?

Lines   192-193: This sentence has no grammatical sense;

This was corrected into (The high antibacterial activity of the protein components (7S &   11S) and cip may give the ground to formulate antibacterial drugs with   certain proportion of the protein replacing the antibiotic. This may open the door for new   therapeutic strategy with less use of the synthetic antibiotics in accordance   with [2])

Line   208: “reduced significantly” – where is the statistical analysis described?

There is no statistical analysis   so we delete " Significantly"

Lines   209-211: This conclusion does not make sense the way it is.

This paragraph   was modified to be clearer into (Storage of minced meat with 11S globulin for   15 days at 4°C reduced significantly the growth of total viable count,   psychrotrophic bacteria and coliforms as compared to controls. This is a promising   result to use 11S   globulin of cowpea as a meat additive in processed meat products.)

“Conclusion” section:

Line   382 – “relatively safer” – than what? A comparison has been made?

Line   382 – “relatively low price” – Compared to what?

This   part was modified into (particularly when they are safe natural products and can be   prepared at considerably low costs)

Lines   382-383: “rich nutritional value” – Was this evaluated? This is not a   conclusion from this study.

This was   modified into: apart from their known rich nutritional   character as legume proteins.

“References”   should be carefully revised in terms of spelling, spacing, journal   abbreviations, italics, etc. There are many errors.

Done-   References   were revised in all the manuscript.

Reviewer 3 Report

Minor remarks:

1.      Lines 34, 37 and 43 – please note the range of refs (for instance, 1-5 and not 1,2,….)

2.      Please cancel bold style for all the refs

3.      Please specify abbreviations at the 1st mention. Line 62 CNP – specified on line 235, line 95 cip – specified on line 300, line 176 IZD – specified on line 289

4.      Lines 82-83 – please supply refs for interpretation of IR bands

5.      Line 80 – please close parentheses after C=O

6.      Line 84 – cm-1, please

7.      Line 85 – what do you mean by “difference spectra”?

8.      Line 87 – please title this section as “Antibacterial properties of …” and not MIC of … - in this section you describe not only MIC, but a disc diffusion method as well

9.      Line 93 - lease add spaces between 50 and unit

10.   Table 1 – K. pneumonia, please

11.   Fig 3 – P. aeruginosa, please – in the top of the Fig and in the caption

12.   Please be consistent in presentation of units – or µg mL-1 (this presentation is preferable) or µg/mL, but not once the former, once the latter

13.   Line160 – “due to variant effect” – what do you mean?

14.   Line 300 – ciprofloxacin – spelling

15.   Line 229 – “provided” – why it is bold and in a wrong font size?

16.   Lines 351-353 – why part of the text is in a bold font?

17.   References. Most of the refs do not correspond to the Journal rules – please fix the refs according to instructions. Please pay special attention on refs 1, 5, 21, 63, 65, 67 – but not only – there is a lot of problematic presentations.

Author Response

Reviewer   # 3

1.      Lines   34, 37 and 43 – please note the range of refs (for instance, 1-5 and not   1,2,….)

Corrected

2.      Please   cancel bold style for all the refs

References were   revised in all the manuscript.

3.      Please   specify abbreviations at the 1st mention. Line 62 CNP –   specified on line 235, line 95 cip – specified on line 300, line 176 IZD –   specified on line 289

Done

4.      Lines   82-83 – please supply refs for interpretation of IR bands

There is ref. 47

5.      Line   80 – please close parentheses after C=O

Done (C= O).

6.      Line   84 – cm-1, please

Done and the Figure was ameliorated.

7.      Line   85 – what do you mean by “difference spectra”?

It is (different spectra).

8.      Line   87 – please title this section as “Antibacterial properties of …” and not MIC   of … - in this section you describe not only MIC, but a disc diffusion method   as well

Done

9.      Line   93 - lease add spaces between 50 and unit

Done

10.   Table   1 – K. pneumonia,   please

Done

11.   Fig   3 – P. aeruginosa,   please – in the top of the Fig and in the caption

Done

12.   Please   be consistent in presentation of units – or µg mL-1 (this   presentation is preferable) or µg/mL, but not once the former, once the   latter

Done

13.   Line160   – “due to variant effect” – what do you mean?

Corrected into: (may be due to different plant varieties)

14.   Line   300 – ciprofloxacin – spelling

Done

15.   Line   229 – “provided” – why it is bold and in a wrong font size?

Done

16.   Lines   351-353 – why part of the text is in a bold font?

Done

17.   References.   Most of the refs do not correspond to the Journal rules – please fix the refs   according to instructions. Please pay special attention on refs 1, 5, 21, 63,   65, 67 – but not only – there is a lot of problematic presentations.

   Done- References were revised in all the manuscript.

Round  2

Reviewer 1 Report

The revisions have been properly made, so the paper can be published

Author Response

Thank you.

Reviewer 2 Report

Manuscript ID: molecules-457505

Characterization and antibacterial activity of 7S and 11S globulins isolated from cowpea seed protein

Abdel-Shafi and co-authors have provided a revised version of the above-mentioned manuscript, following peer-review. The manuscript has improved in several aspects as in the language grammar and spelling, in the logical sequence of the introduction section, in the details in methods and results sections, in the accuracy of bibliographic references. Some minor corrections to the text are still needed before acceptance for publication. Please see below.

Minor corrections:

1.         Table 1) Please replace the following: “S. typhi” by “S. typhimurium”; “St. pyogenes” by “S. pyogenes”; “P. auriginosa” by “P. aeruginosa”; “K. pneumonia” by “K. pneumoniae”; “monocytogens” by “monocytogenes”;

2.         Line 99: Correct “L. monocytogens” into “L. monocytogenes”

3.         Line 103: “were used” instead of “are used”

4.         Lines 106-107: “against L. monocytogenes and S. aureus ‘using’ combinations of antibiotic (Cip)“

5.         Lines 133 and 134: please remove the commas (‘’) from both these lines

6.         Lines 139-140: Minced meat samples were inoculated with both concentrations 50 and 100 μg/g – delete “both concentrations”

7.         Line 226: plural - “new therapeutic strategies”

8.         Line 319: “was immediately” instead of “is immediately”

9.         Line 335: (cip) after ciprofloxacin, not before

Author Response

Reviewer   # 2- Round 2#

1.           Table 1) Please replace the following: “S. typhi” by “S. typhimurium”; “St.   pyogenes” by “S. pyogenes”; “P. auriginosa” by “P. aeruginosa”; “K.   pneumonia” by “K. pneumoniae”; “monocytogens” by “monocytogenes”;

Done

2.           Line 99: Correct “L. monocytogens” into “L. monocytogenes”

Done

3.           Line 103: “were used” instead of “are used”

Done

4.           Lines 106-107: “against L. monocytogenes and S. aureus ‘using’ combinations   of antibiotic (Cip)“

Done

5.           Lines 133 and 134: please remove the commas (‘’) from both these lines

Done

6.           Lines 139-140: Minced meat samples were inoculated with both concentrations   50 and 100 μg/g – delete “both concentrations”

Done

7.           Line 226: plural - “new therapeutic strategies”

Done

8.           Line 319: “was immediately” instead of “is immediately”

Done

9.           Line 335: (cip) after ciprofloxacin, not before

Done